# Schistosomiasis amongst adolescent boys in non-lakeshore southern Malawi: Investigating local risk-factors within a nested community-based cross-sectional survey

Oscar Herrera[1]*, Stefan Witek-McManus[1], James Simwanza[2], Lyson Samikwa[3], Stella Kepha[4], Rejoice Msiska[2], Sean Galagan[5], Elliott Rogers[1], Peter Makaula[6], J. Russell Stothard[7], Judd Walson[5], Lazarus Juziwelo[8†], Rachel Pullan[1], Khumbo Kalua[2,3], Robin Bailey[9]

**1** Department of Disease Control, Faculty of Infectious and Tropical Diseases, London School of Hygiene & Tropical Medicine, London, United Kingdom, **2** Blantyre Institute for Community Outreach, Blantyre, Malawi, **3** Kamuzu University of Health Sciences, Blantyre, Malawi, **4** Eastern and Southern Africa Centre of International Parasite Control, Kenya Medical Research Institute, Nairobi, Kenya, **5** The DeWorm3 Project, University of Washington, Seattle, Washington, United States of America, **6** Research for Health Environment and Development, Mangochi, Malawi, **7** Department of Tropical Disease Biology, Liverpool School of Tropical Medicine, Liverpool, United Kingdom, **8** National Schistosomiasis and Soil Transmitted Helminth Control Programme, Community Health Sciences Unit, Ministry of Health and Population, Lilongwe, Malawi, **9** Department of Clinical Research, Faculty of Infectious and Tropical Diseases, London School of Hygiene & Tropical Medicine, London, United Kingdom

† Deceased

\* oscarflherrera@yahoo.co.uk

## Abstract

### Background

Schistosomiasis is endemic to Malawi, where preventive chemotherapy by mass drug administration (MDA) has been the foundational public health strategy for over a decade. Despite ongoing control, our understanding of the contemporary epidemiology of schistosomiasis in rural Malawi is limited to infrequent school-based surveys, typically lacking evidence from community-based surveys particularly within non-lakeshore upland communities who may be perceived to be at lower risk.

### Methods

Between July and August 2022, we conducted a cross-sectional parasitological survey amongst a community-representative sub-sample of boys aged 10-15 years who had been randomly selected and recruited to the DeWorm3 endline survey in Namwera, Mangochi District. A total of 306 participants from 38 communities were assessed for *S. mansoni* by duplicate Kato-Katz thick smears. Of these, 243 (79.4%) subsequently provided a urine sample to be assessed by filtration for *S. haematobium* and 238 (77.8%) responded to a risk-factor questionnaire. A parallel malacological survey of eight locally important water contact sites was conducted.

**Data availability statement:** All relevant data are within the manuscript and its Supporting Information files.

**Funding:** This work was supported by the London School of Hygiene and Tropical Medicine (No grant number provided, to OH) as well as the Gates Foundation (INV-022149 to OH). The funders played no role in study design, data collection and analysis, decision to publish, or preparation of the manuscript.

**Competing interests:** The authors have declared that no competing interests exist.

## Results

The overall prevalence of egg-patent schistosomiasis was 50.6% (95% CI 44.2-57.1). The prevalence of *S. haematobium* was 47.7% (95% CI 41.3-54.2), of which 37.9% (n=44) were heavy intensity infections whereas the prevalence of *S. mansoni* was 6.5% (95% CI 4.0-9.9), with one moderate intensity infection (0.3%). There was strong evidence of a positive association between detected *S. haematobium* infection and reporting "red urine" (p<0.001) and 'bilharzia' (p=0.005). *Biomphalaria* spp. were found at two sites while *Bulinus* spp. were found at five sites.

## Conclusion

Despite multiple years of MDA at reportedly high coverage, we observed a high egg-patent prevalence with high prevalence of heavy intensity infections amongst boys aged 10–15 years. This evidences engrained and ongoing transmission requiring additional efforts to gain and sustain effective control. Our findings highlight the importance of epidemiological monitoring alongside a schistosomiasis control programme, particularly in areas historically perceived to be at lower risk.

### Author summary

Schistosomiasis is a neglected tropical disease caused by infection with parasitic flatworms that reside in the blood. Preventive chemotherapy by mass drug administration (MDA) with praziquantel for the control of schistosomiasis has been conducted in Malawi for many years but there is limited evidence as to how successful this has been at reducing the prevalence of infection, particularly in rural upland communities residing in areas historically seen as lower risk. This study investigated the prevalence of schistosomiasis in an area of southern Malawi not located near a large body of water, in addition to exploring local risk factors. To do this, we screened 306 10–15-year-old boys for egg-patent schistosomiasis infection by analysing their stool and urine and conducting questionnaires. We found that approximately half the study population was infected with schistosomiasis, and that the prevalence of microscopic blood in participants urine was similar to the prevalence seen via microscopy. We found a higher prevalence of schistosomiasis than recent studies conducted by the shore of Lake Malawi, as well as a prevalence higher than district-level routine monitoring. Further research is required to establish why prevalence of schistosomiasis in this area has remained high despite sustained delivery of MDA. This study demonstrates the critical importance of community based cross-sectional surveys to monitor and evaluate the effectiveness of schistosomiasis control strategies.

## 1. Introduction

Schistosomiasis is a neglected tropical disease caused by infection with parasitic flatworms of the genus *Schistosoma* that reside in the blood. It is estimated that over 238 million people are currently infected with schistosomiasis [1], the majority of whom reside in sub-Saharan Africa where either *Schistosoma mansoni* (intestinal schistosomiasis) or *Schistosoma haematobium* (urogenital schistosomiasis) are endemic [2]. Infection in children is associated with worse school performance, reduced exercise tolerance, malnutrition and anaemia [3–6]. Targeted for elimination as a public health problem by 2030, control of schistosomiasis morbidity relies predominantly on World Health Organization (WHO) recommendations of preventive chemotherapy by annual mass drug administration (MDA) with treatment coverage of ≥75% amongst communities with ≥10% prevalence [7]. Biannual MDA is also recommended if schistosomiasis prevalence is shown to be > 50% amongst school age children (SAC), although evidence demonstrating the effectiveness of biannual MDA is weak and it remains operationally challenging due to global limitations in praziquantel supply [8].

Schistosomiasis is endemic throughout Malawi, where almost half the population is considered at risk [9]. Amongst urine filtration surveys performed in Malawi between 1988 and 2014, a mean prevalence of *S. haematobium* of 23.7% was demonstrated, in the same period Kato-Katz surveys demonstrated a mean prevalence of *S. mansoni* of 7.4% [10]. In 2019, a cross-sectional survey of primary school children in shoreline schools in Mangochi district, approximately 20km away from the study site, demonstrated by urine filtration a *S. haematobium* prevalence of 24% [11]. MDA has been conducted nationally in Malawi since 2009, with the most recent coverage data reporting that in 2019 praziquantel MDA coverage in Mangochi district was estimated to be 75% and 76% for schools and community respectively [12]. Snails of the genus *Bulinus*, the intermediate host species for *S. haematobium*, have long been known to be endemic to Lake Malawi [13]; however snails of the genus *Biomphalaria*, the intermediate host for *S. mansoni*, were not considered endemic to Lake Malawi until as recently as 2017 [14] with parallel studies demonstrating an outbreak of autochthonous transmission of intestinal schistosomiasis [11,15].

Conducting routine schistosomiasis monitoring and evaluation using microscopic methods (i.e. Kato-Katz and urine filtration) is not always feasible due to the cost, equipment or expertise required; therefore, techniques such as symptom questionnaires, Circulating Cathodic Antigen (CCA) and urine reagent strips become more practical [16]. Of all the *Schistosoma* species, questionnaires demonstrate the best performance when detecting *S. haematobium* infection due to the conspicuous and comparatively sensitive nature of macroscopic haematuria with an estimated pooled sensitivity of 0.88 [17]. Increasing prevalence of reported 'red urine' has generally been found to be associated with increasing *S. haematobium* prevalence [18], the WHO red urine study suggested that a prevalence of reported blood in urine ≥30% is equivalent to the prevalence of *S. haematobium* ≥ 50% [19]. The performance of questionnaires as a mapping tool for *S. mansoni* is so variable it is generally recommended that it is validated by parasitological methods before implementation [16]. Urine reagent strips are cheap and widely available, and as such haematuria is often used as a proxy for *S. haematobium* infection, with evidence that a single urine reagent strip is more sensitive than a single urine filtration in certain settings [20].

Rigorous monitoring and evaluation of control strategies such as MDA is key to understanding the effect of such interventions, and in ensuring the efficient use of resources in responding to the changing distribution of schistosomiasis. As such, this study was motivated by the notable lack of contemporary schistosomiasis surveys in this area amongst non-lakeshore upland communities, as well as parasitological findings in the baseline DeWorm3 survey. The primary objective of this study was to estimate the prevalence of both *S. mansoni* and *S. haematobium* infection amongst 10–15-year-old boys in Namwera, an area in the eastern region of Mangochi District in Malawi and to assess water exposure and school attendance as risk factors for schistosomiasis. The secondary objective of this study was to establish the performance of both symptom questionnaire and urine reagent strip as schistosomiasis survey tools. To our knowledge no parasitological or malacological surveys for schistosomiasis have been conducted in Namwera and as such an additional objective was to also establish the presence of intermediate host snail species of schistosomiasis in the area.

## 2. Methods

### 2.1. Study setting

This study was conducted on the Namwera plateau, a rural area in the east of Mangochi district of the Republic of Malawi that is bounded by the Namizumu and Mangochi forest reserves and the Republic of Mozambique. The eponymous town of Namwera is located 23 kilometres by straight line and 38 kilometres by road from Lake Malawi; and is 800 metres above sea level and 300 metres above Lake Malawi (Fig 1). No large lakes or river exist in Namwera, with most of the waterways consisting of small streams and seasonal wetlands that drain towards Mozambique. Schistosomiasis exhibits seasonality of transmission in Malawi with transmission peaking during the rainy reason (December to March) around inland water bodies [21]. A community census conducted in 2017 in Namwera found that 12.3% of people aged 15 and over had completed primary school and that over half of the population is under the age of 18, with 10–15-year-old boys representing 8% of the total population [22].

This study was nested within the DeWorm3 study, a multi-site cluster-randomised trial that aims to assess the feasibility of interrupting soil transmitted helminth infection (STH) using biannual MDA with albendazole [23,24]. Following a site-wide census, communities (villages) were delineated into 40 study clusters of between 1650 and 4000 individuals. Where possible the study cluster preserved the Health Surveillance Assistant catchment area, with one or more villages combined to form each cluster, and no villages subdivided. Individuals were randomly selected from each cluster and assessed at baseline, six months after the final round of MDA (midline) and 24 months post-MDA (endline) for STH infection by collection of a single stool sample subsequently tested by quantitative polymerase chain reaction. The baseline Kato-Katz survey, conducted in 2018 demonstrated a 0.7% prevalence of *S. mansoni* amongst SAC and six cases of egg patent *S. haematobium* in stool [22]. This nested study was conducted from May to July 2022, during the first two months of the endline survey while 17 of the study clusters were being surveyed (Fig 2).

### 2.2. Ethical considerations

The DeWorm3 trial is registered at ClinicalTrials.gov (NCT03014167) and was approved by the College of Medicine Research Ethics Committee (COMREC) at the University of Malawi (P.04/17/2161), the London School of Hygiene & Tropical Medicine (LSHTM) Observational/Interventions Research Ethics Committee (12013) and the Human Subjects Division at the University of Washington (STUDY00000180). Further ethical approval for this study was granted by LSHTM (27665) and the Malawi Ministry of Health and Population. Consent and assent forms, translated into Chichewa, were administered by study enumerators to the participant's parent or caregiver and the participant respectively. If the participant's parent or caregiver was unable to read or write, an impartial witness was identified to complete the consent form on their behalf. No incentives were offered to participate beyond those already offered during the parent trial. Detected parasitic infections were treated by the relevant Health Surveillance Assistant as per WHO guidelines.

### 2.3. Sample size

This study's sample size was determined based on the expected prevalence of microscopic haematuria. Participants 10–15 years old were chosen as they generally exhibit the heaviest infections and highest prevalence of the community, in part due to frequent water exposure [13], with boys specifically as microscopic haematuria in principle should demonstrate the best positive predictive value in this group as haematuria of other aetiologies is less common (such as menstruation or neoplasm). The estimated population of 10- to 15-year-old boys in this sub-section of the DeWorm3 study area was 1633 (Fig 2). The prevalence of *S. haematobium* was assumed to be 24%, based on a recent nearby school based parasitological survey [11], and it was assumed a prevalence of *S. haematobium* of 24% would translate to an identical prevalence of microhaematuria [25]. Using a confidence limit (absolute precision) of 5.5%, a design effect of 1, and a confidence level of

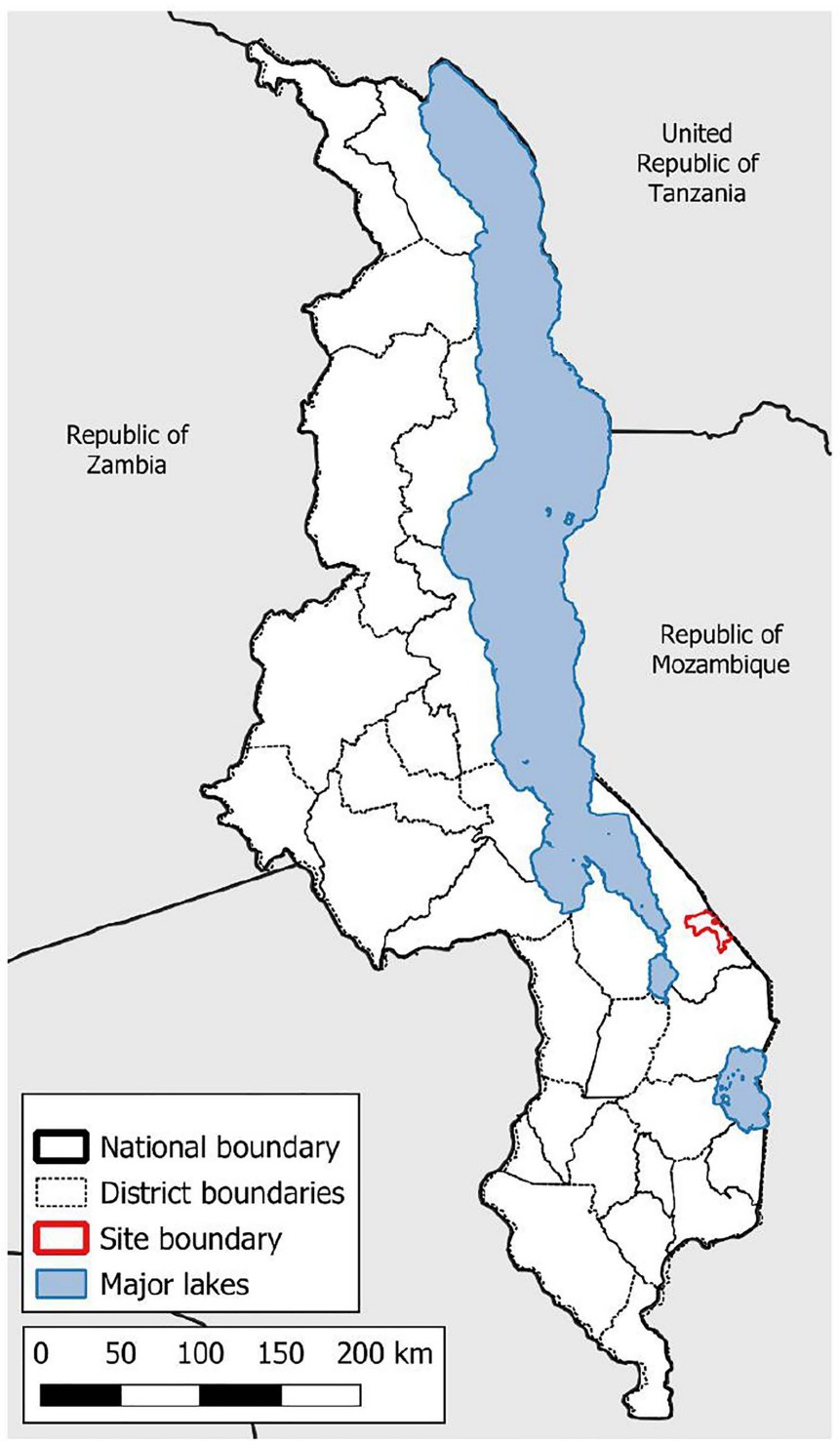

**Fig 1. Location of study site within Malawi.** Contains information from Natural Earth and OpenStreetMap [https://www.naturalearthdata.com/ http//www.naturalearthdata.com/download/10m/cultural/ne_10m_admin_0_countries.zip] [https://www.naturalearthdata.com/ http//www.naturalearthdata.com/download/10m/physical/ne_10m_lakes.zip] Terms of use; Natural Earth: [https://www.naturalearthdata.com/about/terms-of-use/], Open Street Map: [https://www.openstreetmap.org/copyright].

PLOS Neglected Tropical Diseases

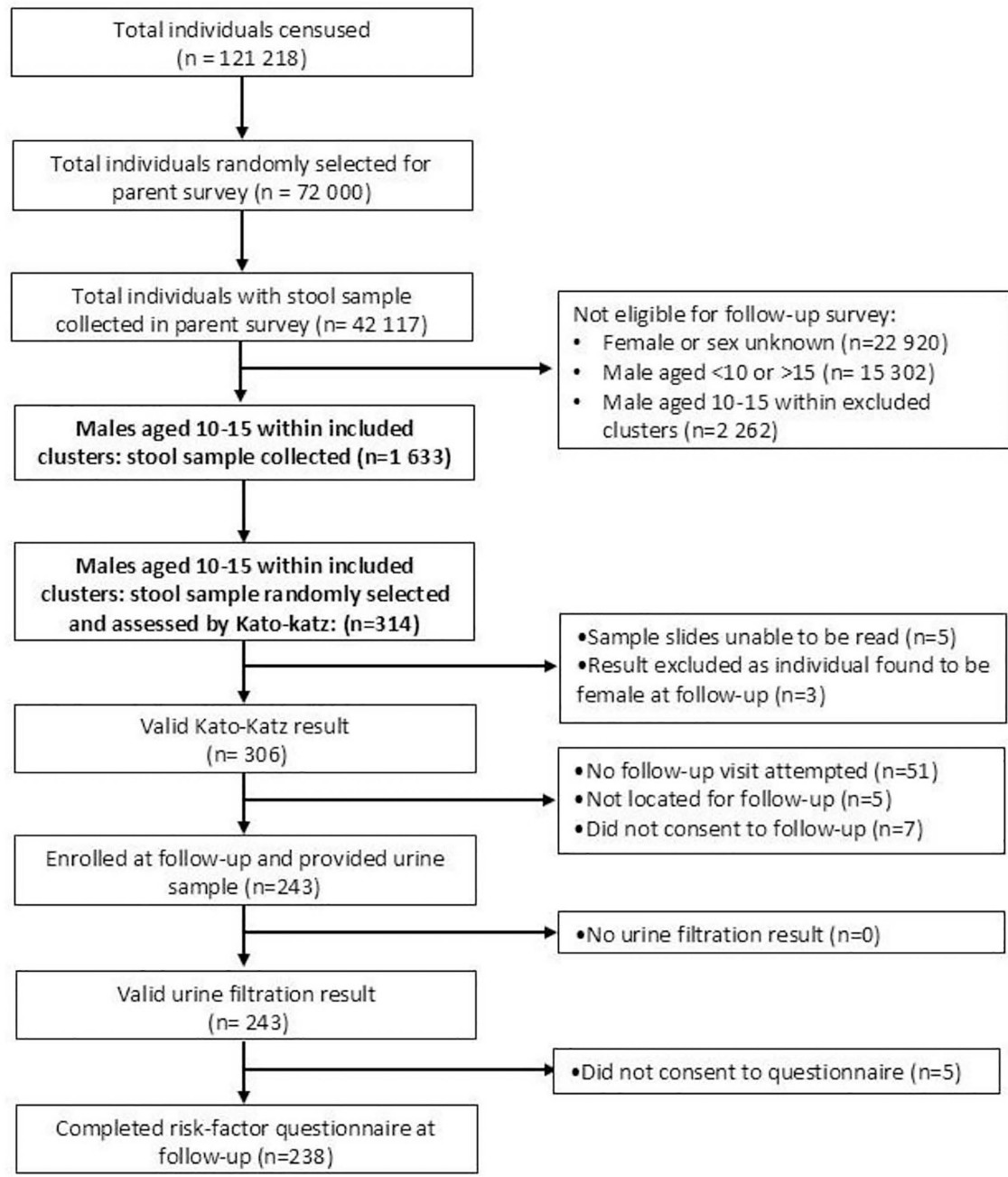

**Fig 2. Study Flowchart.**

95%, a sample size of 203 participants was required. This was calculated using OpenEpi (Sample Size for Frequency in a Population) [26].

## 2.4. Parasitological and urine assessments

Stool samples collected from participants' homes by the DeWorm3 study were refrigerated overnight and aliquoted for qPCR the following morning. The stool remaining after aliquoting was used for Kato-Katz in this study. At the start of each

day where Kato-Katz was performed, a sampling list of eligible stool samples collected the previous day was generated in Stata 17 (College Station, TX: StataCorp LLC) and samples randomly ranked within study cluster. From this list, a maximum number was set for how many samples could be analysed from each study cluster according to laboratory capacity. In line with parent study protocols and WHO recommendations, two slides were prepared and read from each sample at 30–60 minutes after preparation, both slides were read a second time after 18–24 hours to better visualise *Schistosoma* spp eggs [27,28]. Unreadable slides were discarded when calculating infection intensity. Participants with at least one readable slide were considered to have a valid Kato-Katz result.

Terminal urine samples were collected approximately 2 weeks after stool due to delays in obtaining equipment. However, not all participants were able to be revisited for urine samples due to limitations in transportation. Immediately following collection, urine samples were placed in a cooler box and transported to the lab for analysis. Urine filtration was then performed; 10ml of room temperature urine was passed through Sterlitech polycarbonate 12-micron Membrane Filters. The filters were then placed on slides to be read using compound light microscope for egg count per 10ml of urine. All visualised eggs were assumed to be *S. haematobium*. Syringes were washed thoroughly between every use and filters were not reused.

Urine was dipped using ALLTEST10 Parameter Urine Testing Strips. After dipping, excess urine was wiped away using a paper towel, and the urine reagent strip was left for one minute before being compared to the standardised colorimetric scale. Urine was examined visually for subjective macroscopic haematuria, then all parameters of the reagent strip were noted. Participants were deemed proteinuria positive if ≥ 15mg/dl, haematuria positive if ≥ '+' and leukocyte positive if ≥ 75 leu/µl.

## 2.5. Symptom and water exposure survey

A questionnaire was administered by an enumerator at the time of urine sample collection (S1 File). The symptom questionnaire was based on that used by the Red Urine Study Group [19] and the water exposure questionnaire was based on those used in similar surveys [29,30]. Participants were asked if they were currently suffering from either "bilharzia", "blood in urine" or "blood in stool"; how many full days of school they had attended in the last week; and if they had engaged in any of the following activities in a body of water the past week; 'play', 'fishing', 'bathing', 'washing dishes or clothes' and 'swimming'. Water bodies were explicitly defined as streams, lakes, rivers, ponds, irrigation ditches, swamps or dams.

## 2.6. Malacological survey

QGIS v3.42.2 (Open-Source Geospatial Foundation) was used to visualise the location of individuals positive for schistosomiasis by Kato-Katz during the DeWorm3 baseline survey and identify communities where these cases overlapped with possible waterways. Communities were then selected using convenience sampling based on whether the parent study was working in said community at that time. On visiting these communities, a health volunteers' local knowledge was utilised to find waterways where human-water contact was known to occur (Fig 3). Waterways were assessed for size, accessibility, and water flow. Observed evidence of human to water contact and geographical coordinates were recorded (S1 Fig). Snails were collected by one person for one hour at each site using a sieve attached to a rod as well as hand-picking from vegetation using gloves. Waders were worn and any water to skin contact was treated with ethanol. Collected snails were transported in a large sealable container to the laboratory where their species were identified using the Danish Bilharziasis Laboratory Field Guide [31]. Collected snails were rinsed in unchlorinated water to remove debris and sorted by species, snails of species other than *Bulinus* spp. and *Biomphalaria* spp. were discarded.

## 2.7. Statistical analysis

Statistical analysis of the associations of the risk factors (water exposures and school attendance) with *Schistosoma* spp. infection detected by Kato-Katz or urine filtration was performed. Age was deemed to be a confounder a priori and recoded as a categorical variable by years of age. Associations were first explored using descriptive analysis and statistical

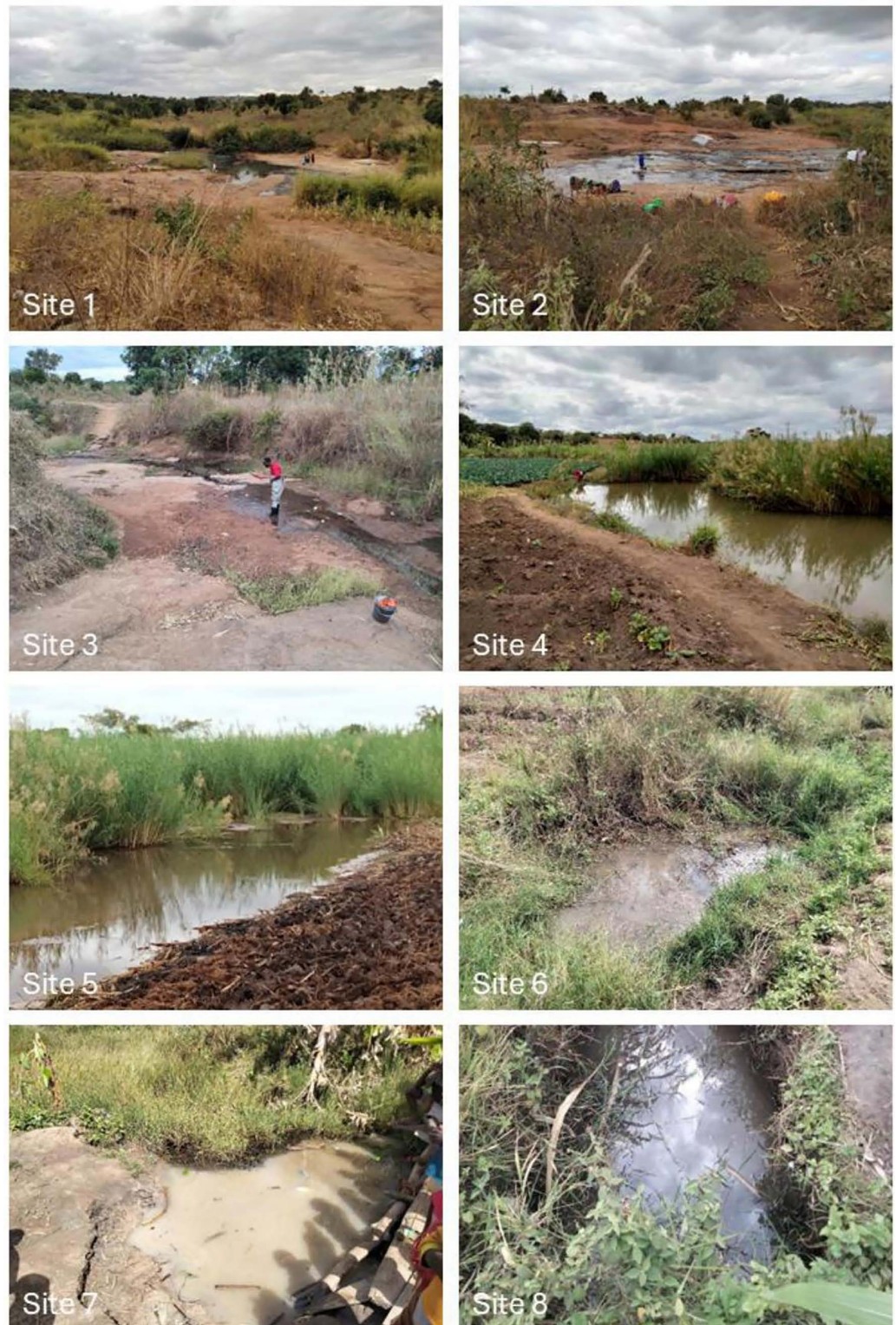

**Fig 3. Photographs of snail survey sites.**

significance was assessed by chi-squared tests. A random effects logistic regression model was constructed including all measured risk factors to model explicitly for between village variation. Odds ratios (ORs) and 95% confidence intervals (CI) were calculated using this model. P-values were calculated using likelihood ratio tests. The model was assessed for data sparsity by ensuring there were at least 10 cases of *Schistosoma* spp. infection for each predictive variable. No interaction terms were fitted as there was likely to be interaction between all measured risk factors and inadequate power. Data was assumed to be missing completely at random. It was assumed that all risk factors combined multiplicatively. Collinearity in this model was assessed by calculating the variance inflation factor score for each variable. The association between *S. haematobium* infection status as a categorical explanatory variable and microhaematuria as a binary outcome was assessed using a random effects logistic regression model adjusted for age as an a priori confounder.

The performance of urine reagent strip parameters was assessed against urine filtration in terms of sensitivity (percentage of infected participants with a positive test result), specificity (percentage of uninfected participants with a negative test result), negative predictive value (NPV = number of uninfected participants with a negative test/ (number of uninfected participants with a negative test + number of infected with a negative test) and positive predictive value (PPV = number of infected participants with a positive test/ (number of infected participants with a positive test+ number of uninfected participants with a positive test) for detecting *S. haematobium* infection. The relationships between schistosomiasis infection status and each symptom questionnaire variable were assessed using Spearman's rank correlation analysis in addition to sensitivity, specificity, NPV and PPV.

All data collection was performed using SurveyCTO (Dobility Inc). Data management and analyses were performed using Stata 17 and maps were constructed using QGIS 3.42.2.

## 3. Results

### 3.1. Participant characteristics

306 participants were included in our final analysis, all of whom were male and between the ages of 10 and 15. The median age was 12.6 years old with an interquartile range of 2.9 years (Fig 4). Participants lived in 38 different villages across 17 study clusters (Fig 5). The number of participants sampled from each cluster ranged from one to 32 with a mean of 18 participants sampled per cluster.

### 3.2. Parasitological assessment

Kato-Katz was performed on the stool samples of 314 participants. In total, 309 (98.4%) samples produced at least one readable slide, five samples (1.6%) were unreadable on all four slides and excluded from our analysis; 10 samples (3.2%) were only readable on one slide, 11 (3.5%) were readable on two slides, 16 (5.1%) were readable on three slides and 272 (86.7%) were readable on all four slides. Three participants initially sampled were later excluded from analysis as they were found to be female at questionnaire delivery.

Of the 306 10–15-year-old male participants with a valid Kato-Katz result, 20 (6.5%, 95% CI: 4.0-9.9) participants had at least one *S. mansoni* egg observed in their stool. 19 participants had light *S. mansoni* infections (< 100 epg), one participant had a moderate infection (100–399 epg), and no participant had a heavy *S. mansoni* infection (≥400 epg). The arithmetic mean epg for those infected with *S. mansoni* was 21.6 epg (95% CI: 8.4-34.8). *S. mansoni* was found in 12 of 15 (80.0%) clusters where at least five participants underwent Kato-Katz, with cluster prevalence ranging from 0.0%-21.4%.

14 (4.6%, 95% CI: 2.5-7.6%) participants were found to have hookworm infection on Kato-Katz. All detected hookworm infections were of light intensity (<2000 epg). One (0.3%) participant had a *Hymenolepis diminuta* infection and 3 (1.0%) participants had light *Ascaris lumbricoides* infections. Five participants (1.6%) had *S. haematobium* infections on Kato-Katz, 3 of those participants underwent urine filtration; one was negative for *S. haematobium*, one had a light infection and the other had a heavy infection.

| Characteristic: | Number of participants n (%)* |
|---|---|
| | n=306 |
| Age (years): | |
| 10 to 11 | 115 (37.6) |
| 12 to 13 | 100 (32.7) |
| 14 to 16 | 91 (29.7) |
| School attendance in past week | |
| None | 55 (23.1) |
| 1 to 4 days | 94 (39.5) |
| 5 days | 89 (37.4) |
| *S. haematobium* infection | |
| Negative | 126 (51.9) |
| Light | 73 (30.0) |
| Heavy | 44 (18.1) |
| *S. mansoni* infection | |
| Negative | 286 (93.5) |
| Light infection | 19 (6.2) |
| Moderate Infection | 1 (0.3) |
| Heavy Infection | 0 (0) |
| Hookworm infection | |
| Negative | 292 (95.4) |
| Light Infection | 14 (4.6) |
| Moderate Infection | 0 (0) |
| Heavy Infection | 0 (0) |

**Fig 4. Study participant characteristics; prevalence and infection intensity of *S. mansoni*, *S. haematobium* and hookworm infection amongst 10- to 15-year-old boys in Namwera, Malawi.** (*) Missing data for school attendance (n = 68) and *S. haematobium* infection (n = 63).

Urine filtration was performed on 243 urine samples with 116 (47.7%, 95% CI: 41.3- 54.2%) samples containing at least one *S. haematobium* egg on filtration. Of the participants that underwent urine filtration, 73 (30.0%) participants had light infections (<50 eggs /10 mL urine) and 44 (18.1%) had heavy infections (≥ 50 eggs /10 mL urine or visible haematuria). The arithmetic mean egg count per 10ml of urine for those infected was 94.0 (95% CI: 64.4-123.5). By cluster where at least five participants underwent urine filtration, *S. haematobium* prevalence ranged from 13.0-77.8%. 9 participants were found to be coinfected with *S. mansoni* and *S. haematobium*.

### 3.3. Urinalysis

Twenty-One (21, 8.6%, 95% CI: 5.4-12.9%) participants had macroscopic haematuria and 127 (52.3%, 95% CI: 45.8-58.7%) participants had greater than trace detectable microscopic haematuria. Additionally, 96 (39.5%) participants had detectable leukocyturia, and 76 (31.3%) participants had detectable proteinuria. Microscopic haematuria on urine reagent strip demonstrated a sensitivity of 89.7% (95% CI: 82.6-94.5), a specificity of 81.9% (95% CI: 74.1-88.2), a PPV of 81.9% (95% CI: 74.1-88.2) and a NPV of 89.7% (95% CI: 82.6-94.5) when compared to urine filtration for detecting *S. haematobium* infection (Fig 6). Sensitivity of microhaematuria by urine reagent strip reading is illustrated in Fig 7. Proteinuria on urine reagent strip demonstrated a sensitivity of 57.8% (95% CI: 48.2-66.9), a specificity of 92.9% (95% CI: 87.0-96.7), a PPV of 88.2% (95% CI: 78.7-94.4) and a NPV of 70.7% (95% CI: 63.1-77.4). Leukocyturia on urine reagent strip demonstrated a sensitivity of 73.3% (95% CI: 64.3-81.1), a specificity of 91.3% (95% CI: 85.0-95.6), a PPV of 88.5% (95% CI: 80.4-94.1) and a NPV of 78.9% (95% CI: 71.4-85.2).

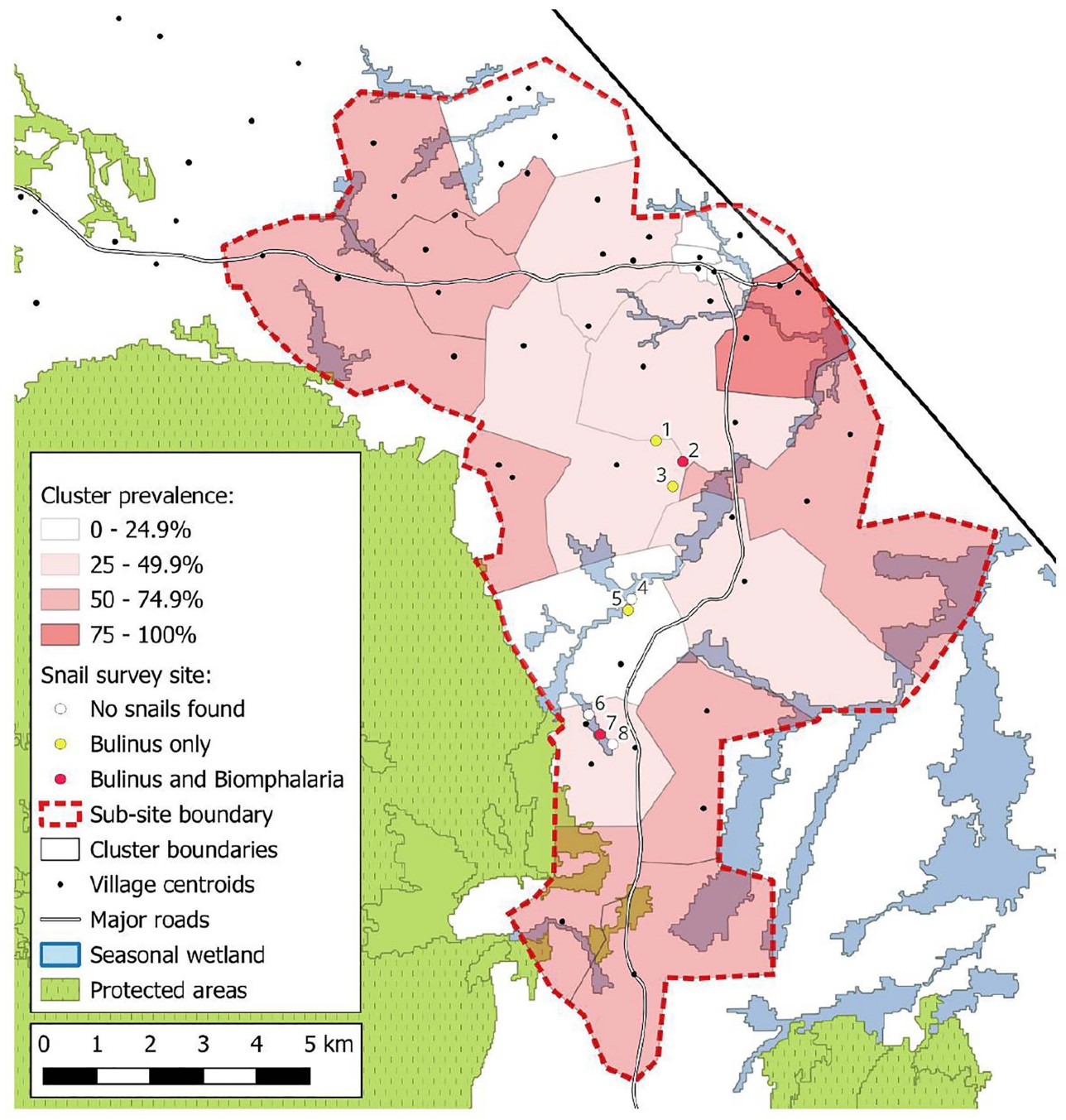

**Fig 5. *S. haematobium* prevalence amongst 10–15-year-old boys by DeWorm3 study cluster with locations of snail survey sites.** Contains information from Natural Earth and OpenStreetMap [https://www.naturalearthdata.com/ http://www.naturalearthdata.com/download/10m/cultural/ne_10m_admin_0_countries.zip] [https://www.naturalearthdata.com/ http://www.naturalearthdata.com/download/10m/physical/ne_10m_lakes.zip]. Terms of use; Natural Earth: [https://www.naturalearthdata.com/about/terms-of-use/], Open Street Map: [https://www.openstreetmap.org/copyright].

| Characteristic | Number of participants n (%) | Number of participants with *S. haematobium* infection n (%) | Sensitivity (95% CI) | Specificity (95% CI) | PPV (95% CI) | NPV (95% CI) | P-value* |
|---|---|---|---|---|---|---|---|
| Haematuria | | | 89.7 (82.6-94.5) | 81.9 (74.1-88.2) | 81.9 (74.1-88.2) | 89.7 (82.6-94.5) | - |
| Negative | 116 (47.7) | 12 (10.30) | | | | | |
| ≥'+' | 127 (52.3) | 104(81.9) | | | | | |
| Proteinuria | | | 57.8 (48.2-66.9) | 92.9 (87.0-96.7) | 88.2 (78.7-94.4) | 70.7 (63.1-77.4) | - |
| Negative | 167(68.7) | 49(29.3) | | | | | |
| ≥15mg/dl | 76(31.3) | 67(88.2) | | | | | |
| Leukocyturia | | | 73.3 (64.3-81.1) | 91.3 (85.0-95.6) | 88.5 (80.4-94.1) | 78.9 (71.4-85.2) | - |
| Negative | 147(60.5) | 31(21.1) | | | | | |
| ≥70 leu/µl | 96(39.5) | 85(88.5) | | | | | |
| "Bilharzia"** | | | 62.9 (52.0-72.9) | 57.3 (47.2-67.0) | 56.0 (45.7-65.9) | 64.1 (53.5-73.9) | 0.005 |
| Yes | 100(42.0) | 56(56.0) | | | | | |
| No | 92(38.7) | 33(35.9) | | | | | |
| Don't know | 46(19.3) | 24(52.2) | | | | | |
| "Red Urine" | | | 64.6 (55.0-73.4) | 67.2 (58.2-75.3) | 64.0 (54.5-72.8) | 67.7 (58.8-75.9) | <0.001 |
| Yes | 114(47.9) | 73(64.0) | | | | | |
| No | 124(52.1) | 40(32.3) | | | | | |

**Fig 6. Diagnostic performance of urine reagent strip parameters and symptom questionnaire compared to urine filtration for the detection of *S. haematobium* infection amongst 10- to 15-year-old boys in Namwera, Malawi.** (*) P-value calculated for symptoms only using spearman rank. (**) Sensitivity, specificity, PPV, NPV and P-value for 'Bilharzia' was calculated by treating 'Don't know' responses as missing data.

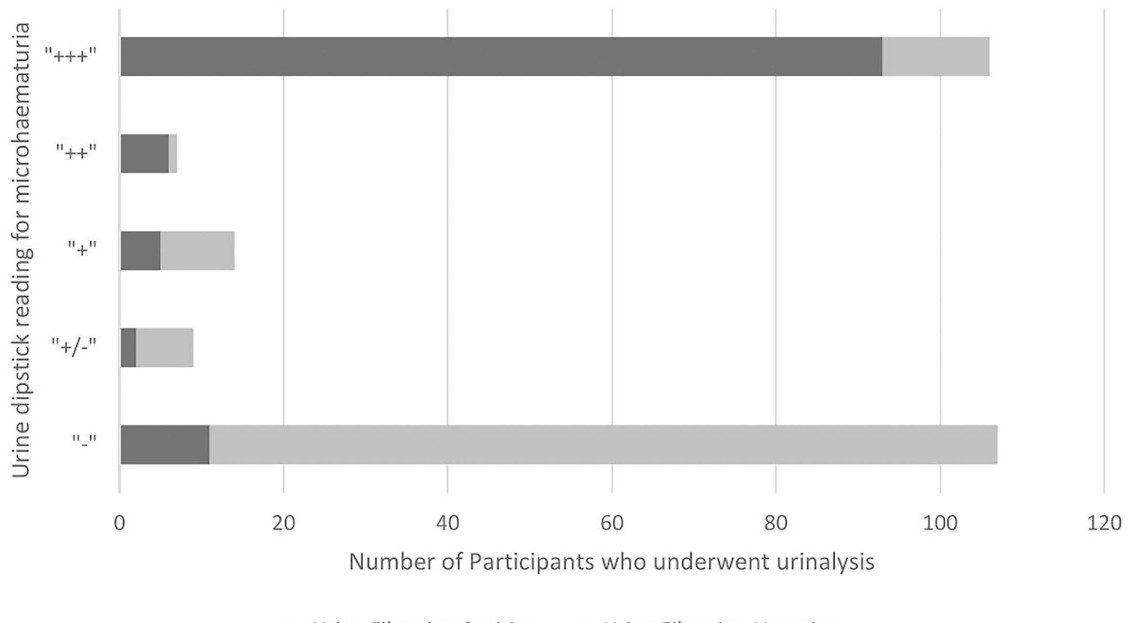

**Fig 7. Urine reagent strip reading by *S. haematobium* infection status in 10- to 15-year-old boys in Namwera, Malawi.**

Using a random effects logistic regression model adjusted for age as an a priori confounder; those with light *S. haematobium* infection on urine filtration had greater odds of detectable microhaematuria in their urine than uninfected participants (AOR: 28.4, 95% CI: 12.5-64.9), with heavy infection demonstrating even greater odds (AOR: 204.7, 95% CI: 26.5-1482.9).

### 3.4. Symptom questionnaire

238 participants completed the questionnaire, provided a valid Kato-Katz result and had their urine analysed. 100 (42.0%) participants believed they currently had "bilharzia", with the most prevalent symptom being "red urine" (n = 114, 47.9%), followed by "blood in stool" (n = 32, 13.5%) (Fig 8).

When "Don't know" was considered missing data, reported "bilharzia" demonstrated a sensitivity of 62.9% (95% CI: 52.0-72.9), specificity 57.3% (95% CI: 47.2-67.0), PPV of 56.0% (95% CI: 45.7-65.9), NPV of 64.1% (95% CI: 53.5-73.9) and a spearman rank p-value of 0.005 compared to urine filtration in the detection of *S. haematobium* infection. "Red urine" demonstrated a sensitivity of 64.6% (95% CI: 55.0-73.4), specificity of 67.2% (95% CI: 58.2-75.3), PPV of 64.0% (95% CI: 54.5-72.8), NPV of 67.7% (95% CI: 58.8-75.9) and a spearman rank p-value of <0.0001.

| Factor | Number of participants n (%) | Number of participants with *Schistosoma* spp. infection n (%) | OR (95% CI)* | AOR (95% CI)† | P-value‡ |
|---|---|---|---|---|---|
| **Swimming** | | | | | |
| No | 155 (65.1) | 83 (53.5) | 1 | 1 | |
| Yes | 83 (34.9) | 37 (44.6) | 0.70 (0.41-1.20) | 0.78 (0.29-2.11) | 0.62 |
| **Bathing** | | | | | |
| No | 150 (63.0) | 78 (52.0) | 1 | 1 | |
| Yes | 88 (37.0) | 42 (47.7) | 0.84 (0.50-1.43) | 3.19 (1.02-9.97) | 0.04 |
| **Fishing** | | | | | |
| No | 200 (84.0) | 103 (51.5) | 1 | 1 | |
| Yes | 38 (16.0) | 17 (44.7) | 0.76 (0.38-1.53) | 0.86 (0.34-2.19) | 0.75 |
| **Washing dishes and clothes** | | | | | |
| No | 143 (60.1) | 78 (54.5) | 1 | 1 | |
| Yes | 95 (39.9) | 42 (44.2) | 0.66 (0.39-1.12) | 0.68 (0.29-1.60) | 0.37 |
| **Playing** | | | | | |
| No | 139 (58.4) | 76 (54.7) | 1 | 1 | |
| Yes | 99 (41.6) | 44 (44.4) | 0.66 (0.39-1.12) | 0.56 (0.21-1.52) | 0.25 |
| **Full days of school attended in past week**** | | | | | |
| 0 | 52 (22.4) | 32 (61.5) | 1 | 1 | |
| 1 | 5 (2.2) | 2 (40.0) | 0.42 (0.06-2.80) | 1.65 (0.63-4.32) | |
| 2 | 47 (20.3) | 17 (36.2) | 0.35 (0.15-0.83) | 1.30 (0.45-3.73) | |
| 3 | 15 (6.5) | 7 (46.7) | 0.55 (0.17-1.77) | 2.91 (0.97-8.74) | |
| 4 | 26 (11.2) | 16 (61.5) | 1.00 (0.38-2.65) | 0.80 (0.27-2.36) | |
| 5 | 87 (37.5) | 42 (48.3) | 0.67 (0.34-1.33) | 1.01 (0.33-3.10) | 0.28 |

**Fig 8. Associations between *Schistosoma* spp. infection and reported water contact and school attendance in the past 7 days, amongst 10- to 15-year-old boys in Namwera, Malawi.** (*) Odds Ratio (OR) from univariate analysis using Mantel-Haenszel. For attendance a univariable logistic regression model was used to calculate OR. (**) 6 participants did not wish to disclose their school attendance. (†) Adjusted Odds Ratio (AOR) calculated using a random effects logistic regression model adjusted for all other measured water contact variables, school attendance and birth year. (‡) Calculated using likelihood ratio test of random effects logistic regression model.

"Blood in stool" demonstrated a sensitivity of 6.7% (95% CI: 0.2-31.9), a specificity of 83.9% (95% CI: 77.9-88.8), PPV of 3.1% (95% CI: 0.1-16.2), NPV of 92.0% (95% CI: 86.9-95.6) and a spearman rank p-value of 0.34 compared to Kato-Katz in detecting *S. mansoni* infection. "Bilharzia" demonstrated a sensitivity of 53.9% (95% CI: 25.1-80.8), specificity of 48.1% (95% CI: 40.7-55.6), PPV of 6.8% (95% CI: 2.8-13.5), NPV of 93.7% (95% CI: 86.8-97.6) and a spearman rank p-value of 0.93 compared to Kato-Katz in detecting *S. mansoni* infection.

### 3.5. Risk factor assessment

87 (37.5%) participants indicated they attended a full week of school in the previous week, and almost a quarter (n = 52, 22.4%) had attended no full days of school in the past seven days. A likelihood ratio test conducted using our full random effect logistic regression model indicated no evidence of association between school attendance and *Schistosoma* spp. infection (p = 0.28).

The most reported water exposure in the past week was 'playing' (n = 99, 41.6%), followed by 'washing clothes or dishes' (n = 95, 39.9%), 'bathing' (n = 88, 37.0%), 'swimming' (n = 83, 34.9%) then 'fishing' (n = 38, 16.0%). 113 participants (47.5%) reported no water exposure in the past week. On multivariate analysis 'bathing' demonstrated the highest AOR of 3.2 (95% CI: 1.0-10.0, p = 0.04). None of the other water exposures explored demonstrated good evidence of association with schistosomiasis infection (Fig 8). None of our variables in our model exhibited a variance inflation factor score of greater than 10, so were all included in our model.

### 3.6. Malacological survey

A total of eight sites were visited across five study clusters (S1 Fig). At three (37.5%) of the sites no snails were found. *Bulinus* spp. were found at five (62.5%) of the sites and *Biomphalaria* spp. were found at two (25.0%) of the sites. Both species were found at 2 of the sites. A total of 41 *Biomphalaria* spp. and 33 *Bulinus* spp. snails were collected across the sites. At one site (site 7) there were innumerable snails, but only 53 snails were collected due to restrictions in processing capacity. Water readings were only taken at the two *Biomphalaria* spp. sites several weeks after snail collection due to limited access to a digital water tester (S2 Fig).

## 4. Discussion

This community-based study of 306 ten to fifteen-year-old boys in Namwera, Malawi demonstrated a *S. haematobium* prevalence of 47.7% and a *S. mansoni* prevalence of 6.5%. Of those participants who underwent both urine filtration and Kato-Katz, 50.6% demonstrated parasitological evidence of *Schistosoma* spp. infection. The high prevalence of observed macroscopic haematuria (8.6%), microscopic haematuria (52.3%) and heavy infection likely indicate high levels of morbidity associated with *S. haematobium* infection in this population. Microhaematuria on urine reagent strip demonstrated high sensitivity (89.7%) and specificity (81.9%) for detecting *S. haematobium* infection. Given the low sensitivity of both urine filtration and Kato-Katz, we have likely underestimated *Schistosoma* spp. prevalence.

The *S. haematobium* prevalence demonstrated is much higher than anticipated given both that annual MDA has likely occurred in Mangochi district over many years and the non-lakeshore topography of our study setting. One hypothesis is the MDA coverage in our sampled population is low, a recent study in Namwera demonstrated that school based deworming data overestimates true coverage [32]. In addition, this study was conducted relatively soon after the Covid-19 epidemic which likely affected MDA delivery. Another hypothesis is that the ecology of Namwera is particularly conducive to the intermediate host snail species compared to the lakeshore. Environmental factors such as local vegetation coverage, elevation and population density likely influence schistosomiasis epidemiology [33]. Human factors such as the prevalence of recent high risk water contact or inadequate WASH intervention may also be leading to high re-infection rates. Though praziquantel resistance is not reported, it must also be considered whether treatment efficacy is lower than anticipated due to host, parasite, or MDA regimen factors. The *S. mansoni* prevalence of 6.5% found in this population is

expected as the mean prevalence demonstrated on Kato-Katz surveys in Malawi is 7.4% [10]. However, the *S. mansoni* prevalence demonstrated is much higher than that of the DeWorm3 baseline survey. Kato-Katz is known to be prone to missing the lowest intensity infections, and in most of our positive slides only one egg was seen. Additionally, there is evidence that *S. mansoni* prevalence is rising in parts of Malawi, perhaps in part due to changing ecology [11]. Any future *S. mansoni* surveys in this population would benefit from use of CCA given the low egg counts and low prevalence demonstrated. The five cases of *S. haematobium* demonstrated on Kato-Katz likely represent ectopic infection due to 'spill-over' of heavy infection or heterospecific mating. All *Schistosoma* spp. eggs in urine were assumed to be *S. haematobium*, it is possible ectopic *S. mansoni* infections were present [34].

The diagnostic performance of urine reagent strip demonstrated is consistent with previous literature and provides evidence that a survey consisting of only urine reagent strip would provide an acceptably accurate measurement of *S. haematobium* prevalence in this moderate-high prevalence community of 10–15-year-old boys. Though it should be acknowledged our study population was, by design, ideal for urine reagent strip diagnostic performance. In contrast, the poor performance of our symptom questionnaire to identify *S. mansoni* infection is not surprising, especially given the low prevalence demonstrated and the co-existence of hookworm infection. As such we would not recommend symptom questionnaires are used to further survey *S. mansoni* in this sub-population.

It is likely the sensitivity of our symptom questionnaire was reduced by our lack of blinding the participants and enumerators to the purpose of the survey. Many people, including school age children, are aware of the symptoms of schistosomiasis. Parents were generally present during the questionnaire and anecdotally many expressed desires for their child to be treated. As such it is possible this cohort over-reported symptoms. Indeed, in this study the percentage of people reporting blood in urine is very similar to *S. haematobium* prevalence, in similar studies questionnaires underestimated prevalence by as much as 20% [35]. In our study both the 'red urine' and 'bilharzia' question showed reasonable specificity and sensitivity, though significantly less than urine reagent strip. If urine reagent strip proves too expensive or is unavailable, then a symptom survey may be a feasible alternative in this population and would have provided a reasonable estimate of prevalence.

The lack of evidence of association between school attendance and schistosomiasis infection was surprising given MDA is almost always distributed on school premises. Our questionnaire only asked about attendance in the past week; it is common for children in Namwera to not attend school for a longer period but then return. Additionally, definitions of 'school' vary, especially in Namwera where children attend both formal (government) education providers and informal institutions which provide predominantly religious education. The latter being far less likely to distribute MDA.

This study has for the first time established the existence of the intermediate host species for both *S. mansoni* and *S. haematobium* in Namwera. We have also demonstrated high frequency water contact in this population, with over half of participants reporting some high-risk water exposure in the past week. It is reasonable to assume that schistosomiasis transmission in this community is autochthonous given how rarely many participants leave their localities, however definitive evidence of viable cercarial shedding in local snails or molecular diagnostics is still required. A more thorough malacological survey is necessary to establish snail density trends and infection prevalence in this area and evaluate the need for snail control programmes.

The parasitological assessments utilised in this survey are subject to several limitations. Given that all but one detected *S. mansoni* infections were of light intensity this study likely would have provided a more accurate prevalence if Kato-Katz had been conducted on multiple stool samples from each participant over several days. Additionally, almost all Kato-Katz readings were performed by one technician, without a formal quality control or blinding process in place. Given these limitations CCA may have been a more appropriate diagnostic method in this population. Most urine samples were provided by the participants immediately within the hours of 10am-2pm, but occasionally the urine pot was left with the participant to collect the first urine of the next morning. The delay between sample production by the participant and analysis varied from 2 to 12 hours. This delay between urine sample production and filtration almost certainly caused an underestimate of

the intensity of infections. Additionally, a quarter of detected *S. haematobium* infection were 'ultra-light' (1–5 eggs per 10ml urine), which are easily missed on microscopy.

This study benefitted from being nested within a community-wide survey with comprehensive census data, unlike most schistosomiasis SAC surveys. Additionally, very few parent study participants did not consent to our follow up, likely due to existing community engagement. Another strength of this study is the comparatively large number of diagnostic/ predictive variables examined, permitting examination of the 'best' survey methods in this population.

There are a number of sources of potential selection bias in our study. Given Kato-Katz and urine sampling only occurred for a limited time within a larger survey, study clusters were unevenly sampled. Additionally, only 79% of participants were revisited to obtain urine due to resource constraints. Compared to other water exposure surveys, the questionnaire used in this survey was very brief and prone to measurement bias. In other surveys participants are often asked about length of exposure, surface area of body exposure and recall exposure over a longer period. Despite this, some weak evidence was found of a positive association between reported bathing in a water body in the past week and *Schistosoma* spp*.* infection. Given that there are likely many confounders not controlled for and comparatively low accuracy of measurement, this evidence should be interpreted cautiously. A more comprehensive water exposure survey in combination with DeWorm3 census data would serve well to inform investment in water, sanitation and hygiene (WASH) interventions. Importantly, this study is unable to account for seasonality of transmission and by being cross-sectional any inference of causality is limited.

Whilst it is not possible to generalise these results to all SAC in Namwera, this study raises the possibility that the prevalence of *Schistosoma* spp*.* infection amongst SAC in Namwera is high enough (>50%) to justify biannual MDA. Given the likelihood of ongoing MDA in this area, we have concerns that Namwera should be defined as a 'persistent hot spot', defined by the WHO as a community with *Schistosoma* spp*.* prevalence greater than 10% despite two rounds of annual MDA with good coverage. We recommend a low cost, community wide survey using urine reagent strips and CCA is promptly conducted, integrated with a questionnaire regarding MDA coverage. Both a more thorough water exposure and malacological survey are required to inform non-MDA interventions. Our study highlights the need for non-lakeshore communities previously regarded as 'low risk' to be included in control programmes and national surveys. We re-emphasise the importance of investment in epidemiological monitoring alongside schistosomiasis MDA programmes, and that this monitoring can be effectively conducted with low-cost screening methods and with minimal technical staff.

## Supporting information

**S1 File.  Questionnaire structure and translation.**
(DOCX)

**S1 Fig.  Location of snail survey sites against satellite image.** Contains information from World Imagery (Esri, DigitalGlobe, GeoEye, i-cubed, USDA FSA, USGS, AEX, Getmapping, Aerogrid, IGN, IGP, swisstopo, and the GIS User Community). Map image is the intellectual property of Esri and is used herein under license. Copyright 2025 Esri and its licensors. All rights reserved. Map base layer; [https://www.arcgis.com/home/item.html?id=10df2279f9684e4a9f6a7f08fe-bac2a9]. Terms of use; [https://doc.arcgis.com/en/arcgis-online/reference/static-maps.htm], [https://support.esri.com/en-us/knowledge-base/what-is-the-correct-way-to-cite-an-arcgis-online-basema-000012040#:~:text=When%20an%20ArcGIS%20Online%20basemap,task%2C%20or%20application%20being%20used], [https://content.esri.com/arcgisonline/docs/tou_summary.pdf].
(TIF)

**S2 Fig.  Snails captured by site and water readings.** (*) Parts per million.
(TIF)

## Acknowledgments

We would like to thank all the BICO enumerators, health surveillance assistants and drivers who performed data gathering and sample collection. Thanks also go to the volunteers from the local villages who aided in locating waterways. Special thanks go to the BICO laboratory team, administration team and cleaning team. We would like to thank the London School of Hygiene and Tropical Medicine for providing additional funding. Finally, we would like to thank all the community members who participated in this study- *zikomo*.

## Author contributions

**Conceptualization:** Oscar Herrera, Stefan Witek-McManus, Khumbo Kalua, Robin Bailey.

**Data curation:** Oscar Herrera, Stefan Witek-McManus, Sean Galagan, Elliott Rogers.

**Formal analysis:** Oscar Herrera, Stefan Witek-McManus, Robin Bailey.

**Funding acquisition:** James Simwanza, Khumbo Kalua, Robin Bailey.

**Investigation:** Oscar Herrera, James Simwanza, Lyson Samikwa, Stella Kepha, Rejoice Msiska, Peter Makaula.

**Methodology:** Oscar Herrera, Stefan Witek-McManus, Lyson Samikwa, Stella Kepha, Peter Makaula, J. Russell Stothard, Robin Bailey.

**Project administration:** Oscar Herrera, Stefan Witek-McManus, James Simwanza, Lyson Samikwa, Rejoice Msiska, Sean Galagan, Elliott Rogers, Judd Walson, Lazarus Juziwelo, Rachel Pullan, Khumbo Kalua, Robin Bailey.

**Resources:** Oscar Herrera, Stefan Witek-McManus, Lyson Samikwa, Stella Kepha, Peter Makaula, J. Russell Stothard, Khumbo Kalua, Robin Bailey.

**Software:** Stefan Witek-McManus, Sean Galagan, Elliott Rogers.

**Supervision:** Oscar Herrera, Stefan Witek-McManus, James Simwanza, Lyson Samikwa, Stella Kepha, Rejoice Msiska, Elliott Rogers, Khumbo Kalua, Robin Bailey.

**Visualization:** Oscar Herrera, Stefan Witek-McManus.

**Writing – original draft:** Oscar Herrera, Stefan Witek-McManus, Robin Bailey.

**Writing – review & editing:** Oscar Herrera, Stefan Witek-McManus, Sean Galagan, Elliott Rogers, J. Russell Stothard, Khumbo Kalua, Robin Bailey.

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
