## [Decision Letter · Decision Letter 0]

24 Jul 2025

Response to Reviewers
Revised Manuscript with Track Changes
Manuscript

Shaden Kamhawi

co-Editor-in-Chief

Paul Brindley

co-Editor-in-Chief

**Additional Editor Comments:**
**Journal Requirements:**

At this stage, the following Authors/Authors require contributions: Oscar Herrera, Stefan Witek-McManus, James Simwanza, Lyson Samikwa, Stella Kepha, Rejoice Msiska, Sean Galagan, Elliott Rogers, Peter Makaula, J. Russell Stothard, Judd Walson, Lazarus Juziwelo, Rachel Pullan, Khumbo Kalua, and Robin Bailey. Please ensure that the full contributions of each author are acknowledged in the "Add/Edit/Remove Authors" section of our submission form.

- ® on page: 10.

5) We notice that your supplementary figures are uploaded with the file type 'Figure'. Please amend the file type to 'Supporting Information'. Please ensure that each Supporting Information file has a legend listed in the manuscript after the references list.

Potential Copyright Issues:

- Please confirm (a) that you are the photographer of Figure S2., or (b) provide written permission from the photographer to publish the photo(s) under our CC BY 4.0 license.

- Figures 1, 2, and S1. Please (a) provide a direct link to the base layer of the map (i.e., the country or region border shape) and ensure this is also included in the figure legend; and (b) provide a link to the terms of use / license information for the base layer image or shapefile. We cannot publish proprietary or copyrighted maps (e.g. Google Maps, Mapquest) and the terms of use for your map base layer must be compatible with our CC BY 4.0 license.

7) Please ensure that the funders and grant numbers match between the Financial Disclosure field and the Funding Information tab in your submission form. Note that the funders must be provided in the same order in both places as well. State the initials, alongside each funding source, of each author to receive each grant. For example: "This work was supported by the National Institutes of Health (####### to AM; ###### to CJ) and the National Science Foundation (###### to AM).".

**Reviewers' comments:**

**Key Review Criteria Required for Acceptance?**

**Methods:**

-Are the objectives of the study clearly articulated with a clear testable hypothesis stated?

-Is the study design appropriate to address the stated objectives?

-Is the population clearly described and appropriate for the hypothesis being tested?

-Is the sample size sufficient to ensure adequate power to address the hypothesis being tested?

-Were correct statistical analysis used to support conclusions?

-Are there concerns about ethical or regulatory requirements being met?

Reviewer #1: •Robust parasitological methods: Kato-Katz and urine filtration are appropriate and well-established diagnostics for S. mansoni and S. haematobium, respectively.

•Integration of malacological data: This strengthens the ecological relevance of the findings and provides context for ongoing transmission dynamics.

•Clear data association: Clear association between self-reported symptoms (e.g., red urine) and infection supports the reliability of symptom-based surveillance in resource-constrained settings.

Reviewer #2: (No Response)

**Results:**

-Does the analysis presented match the analysis plan?

-Are the results clearly and completely presented?

-Are the figures (Tables, Images) of sufficient quality for clarity?

Reviewer #1: The study revealed a surprisingly high prevalence of S. haematobium (47.7%) and a notable burden of heavy-intensity infections (37.9%), suggesting ongoing transmission despite repeated MDA rounds.

Reviewer #2: (No Response)

**Conclusions:**

-Are the conclusions supported by the data presented?

-Are the limitations of analysis clearly described?

-Do the authors discuss how these data can be helpful to advance our understanding of the topic under study?

-Is public health relevance addressed?

Reviewer #1: Using a community-based cross-sectional design nested within the DeWorm3 endline survey, the authors assessed parasitological burden through Kato-Katz and urine filtration, and evaluated behavioral risk factors via questionnaire. A parallel malacological survey at key water contact sites was also conducted. The study revealed a surprisingly high prevalence of S. haematobium (47.7%) and a notable burden of heavy-intensity infections (37.9%), suggesting ongoing transmission despite repeated MDA rounds.

Reviewer #2: (No Response)

**Editorial and Data Presentation Modifications?**

Reviewer #1: 1. Observed weaknesses/Limitations of the study

•Limited age/sex scope: The study focuses solely on male participants aged 10–15, excluding girls and other age groups who may also bear significant burden.

•Low coverage of stool and urine sampling: Only 243 out of 306 participants provided urine samples, and a slightly smaller number completed the risk factor questionnaire, potentially introducing bias or missingness in the risk analysis.

•No serological or antigen-based diagnostics (e.g., POC-CCA or UCP-LF CAA) were included, which limits sensitivity for light infections and hampers comparability with other community-based mapping studies.

•Cross-sectional design limits inference on causality or seasonality, especially as the survey was conducted over just two months (July–August).

•Limited discussion of potential reasons for high S. haematobium despite MDA, such as water contact behaviour patterns, sanitation access, or treatment adherence.

2.Gaps and Suggestions for Manuscript Improvement

•Expand discussion on MDA limitations: The manuscript should further explore why MDA has not reduced heavy infections, discussing factors like coverage accuracy, treatment efficacy, reinfection rates, or drug delivery strategy.

•Broaden risk-factor analysis: Include multivariable regression results (even if not significant) to show which behaviors or exposures were most associated with infection.

•Clarify malacological survey: More detail is needed on snail infection status, cercarial shedding, or snail density trends—critical to linking human and environmental transmission.

•Consider including females in future work: Gendered differences in water contact and school attendance could influence transmission risk. There are no communities with only males only exclusion of the other gender in any study is already bias driven.

•Strengthen conclusions with policy relevance: Offer concrete recommendations, such as inclusion of non-lakeshore areas in control programs, WASH interventions, or entomological surveillance expansion.

Reviewer #2: (No Response)

**Summary and General Comments:**

Reviewer #1: This study investigates the prevalence and risk factors of schistosomiasis among adolescent boys aged 10–15 years in non-lakeshore, upland communities of Namwera, Mangochi District, Malawi a setting often considered to be at lower risk. Using a community-based cross-sectional design nested within the DeWorm3 endline survey, the authors assessed parasitological burden through Kato-Katz and urine filtration, and evaluated behavioral risk factors via questionnaire. A parallel malacological survey at key water contact sites was also conducted. The study revealed a surprisingly high prevalence of S. haematobium (47.7%) and a notable burden of heavy-intensity infections (37.9%), suggesting ongoing transmission despite repeated MDA rounds.

Reviewer #2: (No Response)

PLOS authors have the option to publish the peer review history of their article (what does this mean? ). If published, this will include your full peer review and any attached files.

**Do you want your identity to be public for this peer review?** For information about this choice, including consent withdrawal, please see our Privacy Policy .

Reviewer #1: No

Reviewer #2: No

**Figure resubmission:****Reproducibility:** To enhance the reproducibility of your results, we recommend that authors of applicable studies deposit laboratory protocols in protocols.io, where a protocol can be assigned its own identifier (DOI) such that it can be cited independently in the future. Additionally, PLOS ONE offers an option to publish peer-reviewed clinical study protocols. Read more information on sharing protocols at https://plos.org/protocols?utm_medium=editorial-email&utm_source=authorletters&utm_campaign=protocols

---

## [Decision Letter · Decision Letter 1]

13 Nov 2025

Dear Dr Herrera,

We are pleased to inform you that your manuscript 'Schistosomiasis amongst adolescent boys in non-lakeshore southern Malawi: Investigating local risk-factors within a nested community-based cross-sectional survey' has been provisionally accepted for publication in PLOS Neglected Tropical Diseases.

Best regards,

Brianna R Beechler, Ph.D., DVM

Academic Editor

Francesca Tamarozzi

Section Editor

Shaden Kamhawi

co-Editor-in-Chief

Paul Brindley

co-Editor-in-Chief

Reviewer's Responses to Questions

**Key Review Criteria Required for Acceptance?**

**Methods**

-Are the objectives of the study clearly articulated with a clear testable hypothesis stated?

-Is the study design appropriate to address the stated objectives?

-Is the population clearly described and appropriate for the hypothesis being tested?

-Is the sample size sufficient to ensure adequate power to address the hypothesis being tested?

-Were correct statistical analysis used to support conclusions?

-Are there concerns about ethical or regulatory requirements being met?

Reviewer #2: (No Response)

**Results**

-Does the analysis presented match the analysis plan?

-Are the results clearly and completely presented?

-Are the figures (Tables, Images) of sufficient quality for clarity?

Reviewer #2: (No Response)

**Conclusions**

-Are the conclusions supported by the data presented?

-Are the limitations of analysis clearly described?

-Do the authors discuss how these data can be helpful to advance our understanding of the topic under study?

-Is public health relevance addressed?

Reviewer #2: (No Response)

**Editorial and Data Presentation Modifications?**

Reviewer #2: (No Response)

**Summary and General Comments**

Reviewer #2: (No Response)

PLOS authors have the option to publish the peer review history of their article (what does this mean? ). If published, this will include your full peer review and any attached files.

**Do you want your identity to be public for this peer review?** For information about this choice, including consent withdrawal, please see our Privacy Policy .

Reviewer #2: **Yes: ** Jean-Baptiste K. Sékré

---

## [Editor Report · Acceptance letter]

Dear Dr Herrera,

We are delighted to inform you that your manuscript, "Schistosomiasis amongst adolescent boys in non-lakeshore southern Malawi: Investigating local risk-factors within a nested community-based cross-sectional survey," has been formally accepted for publication in PLOS Neglected Tropical Diseases.

Best regards,

Shaden Kamhawi

co-Editor-in-Chief

Paul Brindley

co-Editor-in-Chief
